# Metabolic signatures in human follicular fluid identify lysophosphatidylcholine as a predictor of follicular development

Jihong Yang[1], Yangbai Li[1], Suying Li[1], Yan Zhang[1], Ruizhi Feng[2,3], Rui Huang[4], Minjian Chen [4,5,6✉] &
Yun Qian [1,6✉]

In order to investigate the metabolic characteristics of human follicular fluid (FF) and to reveal potential metabolic predictors of follicular development (FD) with clinical implications, we analyzed a total of 452 samples based on a two-stage study design. In the first stage, FF samples from both large follicles (LFs) and matched-small follicles (SFs) of 26 participants were analyzed with wide-spectrum targeted metabolomics. The metabolic signatures were described by multi-omics integration technology including metabolomic data and transcriptomic data. In the second stage, the potential biomarkers of FD were verified using enzyme-linked immunoassay with FF and blood serum from an independent 200 participants. We describe the FF metabolic signatures from ovarian follicles of different developmental stages. Lysophosphatidylcholine (LPC) can be used as a biomarker of FD and ovarian sensitivity, advancing the knowledge of metabolic regulation during FD and offering potential detection and therapeutic targets for follicle and oocyte health improvements in humans.

[1] Reproductive Medical Center of Second Affiliated Hospital of Nanjing Medical University, Nanjing 210011, China. [2] State Key Laboratory of Reproductive Medicine, Nanjing Medical University, Nanjing 211166, China. [3] The Second Affiliated Hospital of Nanjing Medical University, Nanjing 210011, China. [4] Key Laboratory of Modern Toxicology of Ministry of Education, School of Public Health, Nanjing Medical University, Nanjing 211166, China. [5] State Key Laboratory of Reproductive Medicine, Center for Global Health, School of Public Health, Nanjing Medical University, Nanjing 211166, China. [6] These authors jointly supervised this work: Minjian Chen, Yun Qian. ✉email: minjianchen@njmu.edu.cn; qianyun@njmu.edu.cn

Follicular fluid (FF) is a major component of oocyte microenvironment for oocyte growth, follicular maturation, and germ cell-somatic cell communication. It accumulates all metabolisms during oocytes growth[1]. Oocytes and follicular somatic cells work synergistically in ovaries to ensure proper metabolism of carbohydrates, amino acids, and lipids[2]. Therefore, FF metabolites analysis can reflect the metabolic state of follicles and help determine the quality of oocyte. During follicular development (FD), metabolite concentrations in the FF undergo certain drastic alterations regarding differences in fertility between heifers and lactating cows[3]. FD-related metabolite alterations were also found in other species as well, such as, pigs[4]. In addition, pathways and substrates involved in glucose metabolism, energy production, and proteinogenesis were present in higher levels in FF of larger lactating beef cow follicles[5]. It was earlier discovered that elevations in androgen and FF lipid accumulation often increase oxidative stress (OS), which may be a major factor affecting decline in fertility[6]. Recently, transcriptomic information of human FF from mature and immature ovarian follicles has been provided, which advanced the understanding of omics changes for FD in FF[7]. However, the global metabolic signatures of follicle developmental stages and the key predictor of FD in FF are still largely unknown in humans[8].

In this study, we used the wide spectrum targeted metabolomics to analyze metabolic alterations in FF from human ovarian follicles of different stages. Targeted metabolomics covers wide spectrum of metabolites, and has advantage in identifying and quantifying them. It helps disclose novel biomarkers for the major clinical entities[9,10]. Our aim was to elucidate the global metabolite signatures in FF in relation to FD in humans by metabolomic analysis and following multi-omics integration, and to identify the key predictor of FD both in FF and blood serum by linking to clinical epidemiological information. This is of great clinical significance as it will aid in the selection of the important metabolic signatures for further research on mechanisms underlying physiological and pathological process during FD. Metabolites related to FD found in FF may be used in culture medium to intervene the quality and development of oocyte, which will provide important information for improving the level of ART in clinical. Meanwhile, studies have shown that FF vitamin D levels were consistent with serum vitamin D levels, and that serum vitamin D levels were positively correlated with normal fertilization rate[11]. So, further, if the link between FF and serum can be found and that a strong positive correlation of some metabolites in both compartments allowing potential use of it as a less invasive biomarker, which holds the promise for both the novel biomarkers discovery for the diagnosis and therapeutic targets discovery for enhancing follicle and oocyte health in humans.

## Results

**Demographic information**. The participants involved in this study were IVF patients with fertility issues and seeking assisted reproductive technology (ART). FF samples from different follicles sizes for metabolomics were taken from 26 participants (Fig. 1a). The age of the participants in this study was $29.50 \pm 0.62$ years, with a body mass index (BMI) of $21.21 \pm 0.38$ kg/m$^2$. The basal ovarian function, hormone level on the trigger day, hyperovulation condition, and ART results are shown in Fig. 1b–e. See Supplementary Table 1 for details. The relevant clinical epidemiological demographics information for the male spouse of all study participants is presented in Supplementary Table 2.

**Metabolic signatures in FF from follicles during growth progression**. The 52 FF samples, consisting of 26 FF samples from large follicles (LFs) and 26 FF samples from small follicles (SFs),

were selected and divided into two groups for studying the metabolic signatures. Metabolomic analysis, based on the coverage of a wide range of targeted metabolomics, identified 524 metabolites (Supplementary Data 1). PCA and OPLS-DA were used for processing of the metabolomic data in LFs and SFs, and it exhibited satisfactory separation (Fig. 2a, b). The goodness of fit (R2) and predictive ability (Q2) of the OPLS-DA model was 0.889 and 0.688 (Fig. 2c), respectively. There were 127 metabolites with variable importance in projection (VIP) > 1 based on the results of OPLS-DA, and the OPLS-DA S-plot diagram revealed that the closer the metabolites were to the upper right and lower left regions, the difference was enhanced (Fig. 2d). Finally, 116 differential metabolites with fold change $\geq 2$ or $\leq 0.5$, VIP > 1, and $p < 0.05$ were identified in our study according to previous reports[12–15], and the result is displayed in the volcano plot (Supplementary Fig. 1). Figure 2e showed the 40 metabolites with VIP $\geq 2$. Due to the large number of differential metabolites, among the 116 differential metabolites, 37 metabolites with $p < 0.03$ and VIP $\geq 2$ were selected for heatmap display in Fig. 3a and Supplementary Fig. 2, which indicated the most dramatic changes in the metabolome.

Using KEGG database[16], the classification diagram of the 116 differential metabolites is provided in Supplementary Fig. 3. By combining the pathway analysis (Fig. 3b, Supplementary Table 3) and network analysis (Fig. 3c), the importance of lysophosphatidylcholine (LPC) metabolism including glycerophospholipid, choline, and acetylcholine was highlighted in FF during FD.

Based on the visualization of multi-omics integration by MetScape plugin Cytoscape, a large number of differential metabolites and mRNAs were directly connected to form the pathway network (Fig. 4). This was built by inputting both statistical and knowledge-based information, and defined according to KEGG database. According to the number of differential metabolites in their classifications, amino acids, nucleotides, and carbohydrates were the main changed classifications of metabolites in the network, and bile acids, cholines as well as various LPCs were also included in the network. Most levels of amino acids, nucleotides, carbohydrates, and cholines were elevated in SFs, compared to the LFs, while the decrease of a series of LPCs (lysoPC 15:0, lysoPC 17:0, and lysoPC 20:2) was notable in SFs, relative to LFs. Meanwhile, the genes including OLFR (olfactory receptor), TAS2R (taste 2 receptor), and VNNs (vanins) were included in the network, through which we described the metabolic signatures of the human FF from ovarian follicles at different stages by integration of metabolomic and transcriptomic data.

**Association between differential metabolites and clinical epidemiological data**. Since the FF samples of LFs and SFs were collected from the same participant, the ratio of the two metabolite concentrations can reflect the state of the metabolite in one individual. We, therefore, calculated the ratios of the 116 differential metabolites and analyzed the correlation between these ratios (SFs/LFs) and the clinical indicators. For correlations with statistical significances, the correlation coefficients and their subsequent $p$-values are summarized in Fig. 5. Among them, age and antral follicle count (AFC) were correlated with 9 differential metabolites, respectively. Ovarian stimulation parameters including gonadotropin (Gn) dose and Gn duration were both inversely correlated to 11 and 10 differential metabolites, respectively. In addition, PGD2 and PGJ2 were found to be positively correlated with follicle-stimulating hormone (FSH) and negatively correlated with anti-mullerian hormone (AMH), which fully indicates that PGD2 and PGJ2 are two indicators reflecting ovarian function. Four metabolites, L-lactic acid,

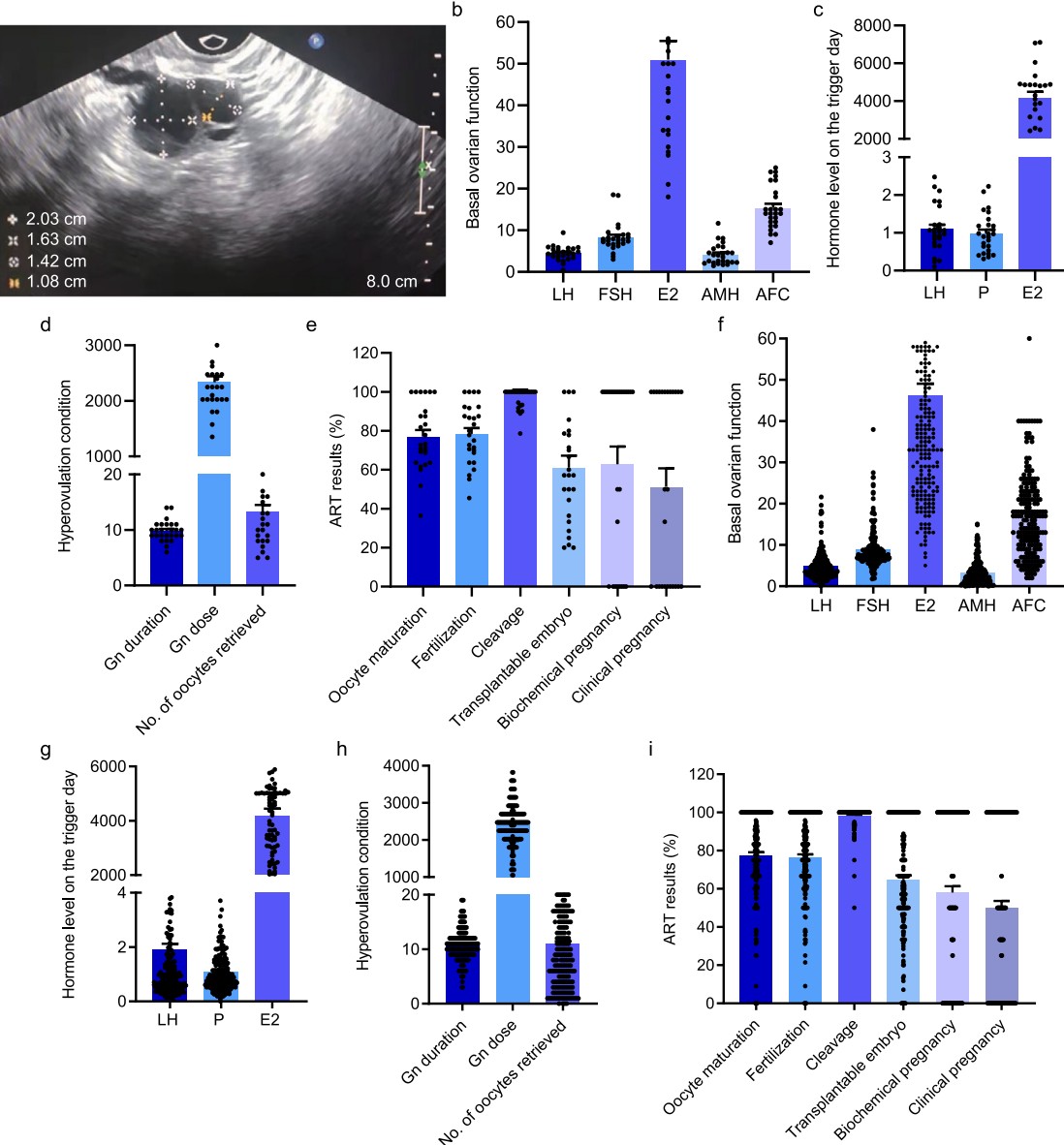

**Fig. 1 The relevant clinical epidemiological information of 26 participants in this study. a** Representative ultrasonic image of large follicles and matched-small follicles. **b** The basal ovarian function included LH (mIU/ml), FSH (mIU/ml), E2 (pg/ml), AMH (ng/ml) and AFC. **c** The levels of LH, P (nmol/L), E2 on the trigger day are presented. **d** Ovarian stimulation parameters (Gn dose and Gn duration) and number of oocytes retrieved could reflect the condition of hyperovulation. **e** ART results included embryonic development and pregnancy results. **f–i** Additional 200 participants' relevant clinical epidemiological information included the basal ovarian function, hormone level on the trigger day, hyperovulation condition, and ART results.

4-methylvaleric acid, 4-hydroxyretinoic acid, and 4-hydroxy-6-methyl-2-pyrone were positively correlated with pregnancy outcome (biochemical and clinical pregnancy rates), whereas myoinositol was negatively correlated with pregnancy outcome. Absence of metabolites was associated with both ovarian function and clinical pregnancy outcome, indicating that there is no direct correlation between ovarian function and clinical pregnancy outcome. Then after FDR correction, we found that the rates of two metabolites PGJ2 and 5-amino-1-[3,4-dihydroxy-5-(hydroxymethyl)oxolan-2-yl]imidazole-4-carboxamide were associated with basal FSH, and the rate of one metabolite Phe-Phe was associated with Gn dose.

**The LPCs in relation to FD and ovarian sensitivity in FF and blood serum**. According to KEGG database, the 116 differential metabolites were divided into 8 classes (Fig. 6a). Based on the

pathway analysis and network analysis results which emphasized the importance of LPC in FF during FD (Fig. 3b and c), we next found that 9, out of 116 differential metabolites, belonged to phospholipids. Therefore, phospholipids occupied a large region of the pathway network (Fig. 4). The 9 phospholipids were downregulated in SFs, except for o-phosphorylethanolamine. Details of these 9 metabolites are shown in Fig. 6b. Next, we combined these 9 metabolites together, and the area under the curve (AUC) was 0.8743 with 76.92% sensitivity and 88.46% specificity (Fig. 6c). For total LPC, the AUC was 0.7426 with 73.08% sensitivity and 69.23% specificity (Fig. 6d).

Based on the above results, we speculated that total LPC may be a good biomarker for further study and potential clinical use. Hence, we collected samples from an independent population of 200 individuals for further validation. Their age was 32.53 ± 0.42 years, with BMI of 23.03 ± 0.27 kg/m$^2$. Figure 1f–i summarized the other relevant clinical epidemiological information, and

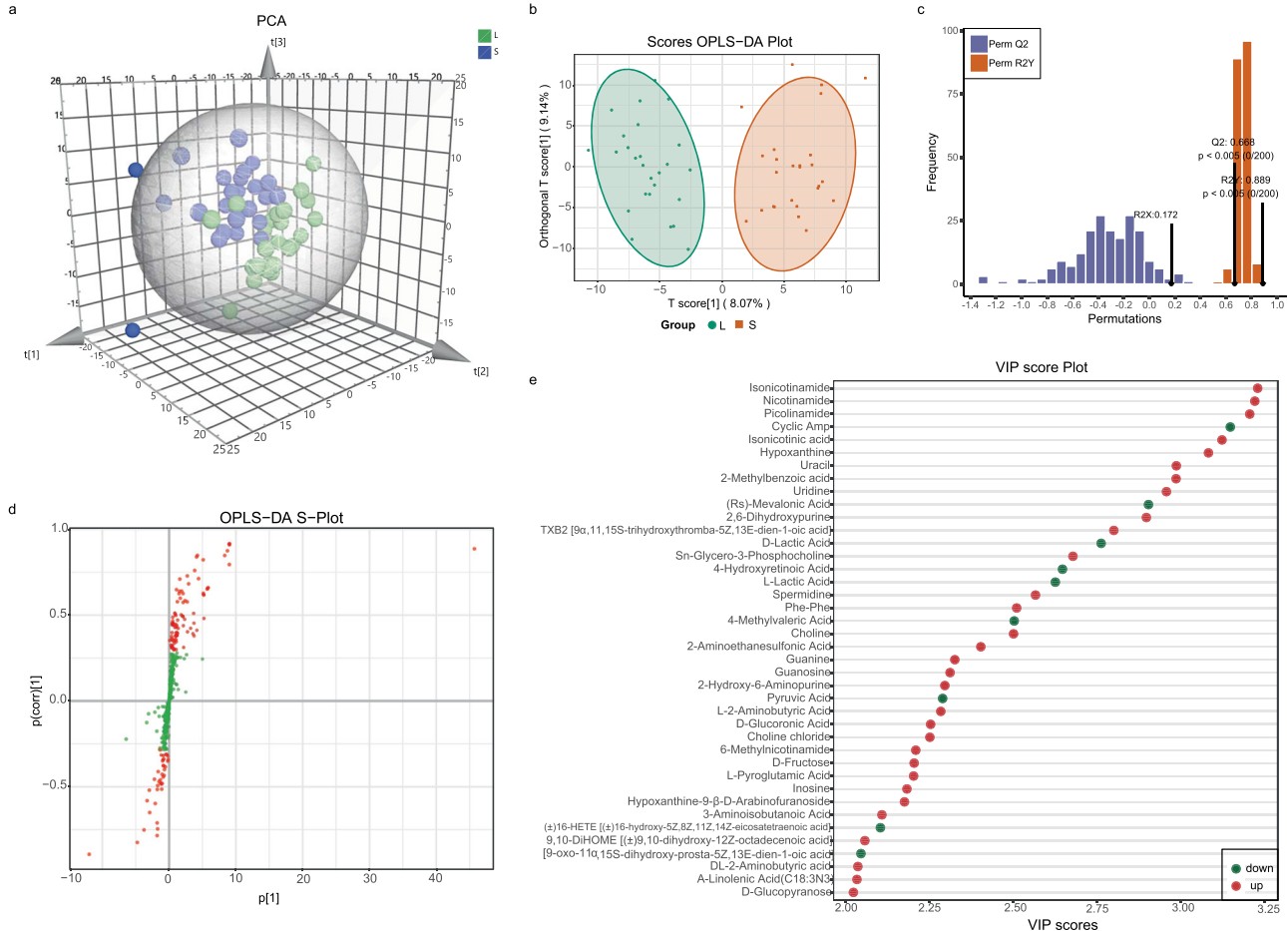

**Fig. 2 The overview of metabolomic analysis of the follicular fluid from human ovarian follicles of different sizes. a** PCA showed the effect of separation between the two groups of follicular fluid samples. **b** OPLS-DA score plot demonstrating good separation between the two groups, based on the regression model. Samples belonging to the large follicles and small follicles are represented by green and red nodes, respectively. **c** The OPLS-DA model validation depicts a prediction power (Q2) of 0.668 and a correlation index (R2Y) of 0.889, based on cross-validation. **d** S-plot of OPLS-DA. Red dot VIP value ≥1, green dot represents metabolite VIP value <1. **e** Alterations in 40 representative differential metabolites with $p < 0.03$ and VIP ≥ 2. Horizontal axis represents the VIP score (contribution of a given variable to the regression model) of each substance.

their male spouses' information was shown in Supplementary Table 4.

After testing the FF and blood serum samples of 200 participants, the total LPC was found to be markedly higher in FF, compared to blood serum (Fig. 6e). In addition, we also observed a strong relationship between the FF LPC and blood serum LPC samples ($r = 0.8383$, $p < 0.001$) (Fig. 6g).

To further determine the relationship between LPC and epidemiological information, correlation analysis was performed. We revealed that LPC in FF was negatively correlated with BMI, AFC, and ovarian stimulation parameters (Gn dose and Gn duration). Additionally, it was positively correlated with ovarian sensitivity parameters (follicular output rate (FORT), follicular sensitivity index (FSI), and follicle-to-oocyte index (FOI)) (Fig. 6f). Moreover, LPC in blood serum was negatively correlated with AFC and positively correlated with ovarian sensitivity parameters (FORT, FSI, and FOI) (Fig. 6f). We observed that blood serum LPC can serve as an effective predictor of ovarian sensitivity parameters FORT, FSI, and FOI, with AUC values of 0.8280, 0.7682, and 0.6496, respectively (Fig. 6h). Meanwhile, the AUC of LPC in FF were 0.7894, 0.7455, and 0.6224 respectively (Fig. 6i). However, we did not find significant associations of LPC concentrations in blood serum and FF with IVF status or pregnancy outcome (Supplementary Table 5 and Supplementary Table 6).

## Discussion

We performed metabolomic analysis of FF to examine potential associations between metabolites and FD. Analyzing both LFs and SFs, we identified 524 metabolites, based on a wide range of targeted metabolomics. The bioinformatics analysis by integrating both metabolomic and transcriptomic data and the FF and blood serum evaluations of 200 additional participants collectively revealed the importance of LPC in FD. The main findings of our study were as follows: (i) FF at different developmental stages exhibited diverse metabolic profiles in humans by targeted metabolomic analysis and following omics integration; and (ii) LPC was closely related to FD and ovarian response.

In this study, we analyzed metabolite levels in LFs and SFs, and observed notable differences in metabolite content during follicular growth progression. It is possible that in the process of follicular enlargement, environmental and physiological factors alter follicular capillary permeability and secretion, which, in turn, affects the metabolite profiles in FF. Among these alterations, a large amounts of metabolites changes were amino acids, nucleotides, carbohydrates, cholines, and various LPCs. The levels of all amino acids, nucleotides, carbohydrates, and choline metabolites were up-regulated in SFs, compared to LFs, while various LPCs, which were down-regulated in SFs, relative to LFs. In rats, FSH alters follicular carbohydrate moiety, which, in turn, promotes carbohydrate-lectin association, which increases

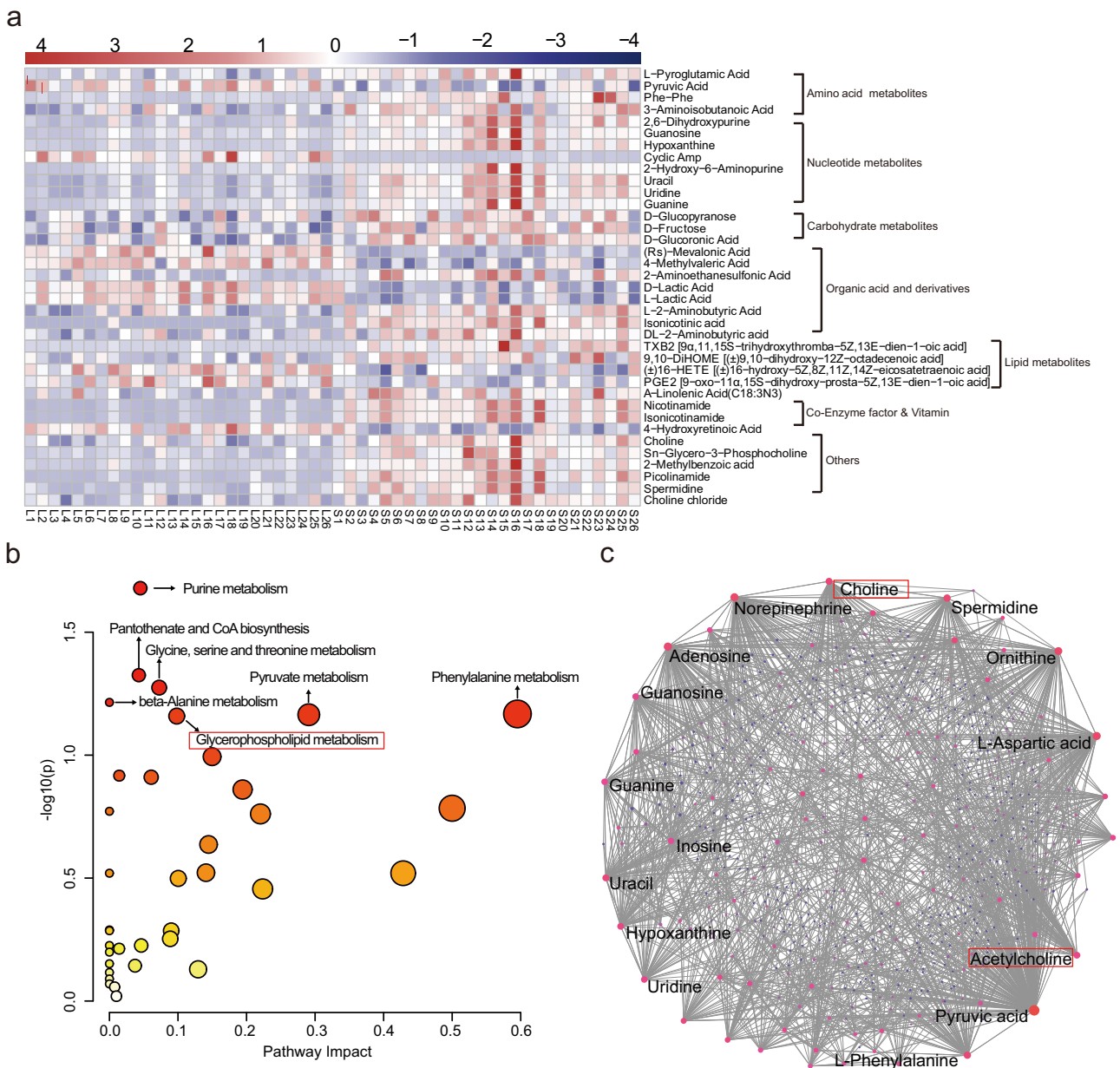

**Fig. 3 Metabolomic pathway analysis of differential metabolites in follicular fluid. a** Heatmap illustrating the representative differential expression of 37 metabolites with VIP ≥ 2 and $p < 0.03$. **b** The pathway analysis results of the 116 differential metabolites. **c** Network analysis results of the 116 differential metabolites.

granulosa cell death. Based on these evidences, carbohydrate is critical for follicular selection and atresia[17]. Studies revealed that metabolism in porcine atresia follicles is characterized by reduced lipids (particularly, phospholipids) content and elevated amino and bile acids. As such, amino and cholic acids dysregulation can result in porcine follicular atresia[18]. These results suggest potential similarity of some key metabolic changes during FD between humans and other mammals. Among the differential metabolites, we found that LPC, a type of phospholipids, was a key biomarker. Lipid is the general term for oils, fats, and lipids, which provides humans with energy and essential fatty acids. Studies have shown that sphingolipids levels in peritoneal fluid are significantly elevated in infertile women with severe endometriosis[19]. Likewise, lipids also encourage oocyte growth and development, aid in oocyte membrane synthesis, as well as modulate cell cycle and survival, as well as malignant

transformation, and apoptosis[20,21]. Hence, lipid metabolism is currently one of the most frequently studied areas of reproductive and developmental biology. Multiple studies demonstrated that the human ovarian follicular environment surrounding the oocyte exhibits a unique metabolite profile, with distinct localisation of lipids within FF and oocytes, and maturation in an environment with altered lipid content may be detrimental to oocytes[22,23]. Balanced amount of various fatty acid concentration is required for optimal oocytes growth and development, for example, higher concentration of saturated fatty acids affects the post-fertilization developmental competence of in vitro matured oocytes, while monounsaturated fatty acids like oleic acid initiate the normal developmental competence[24]. Bertevello et al. demonstrated that alterations in FF lipid composition were associated with follicular growth and oocyte development[25]. Moreover, a prospective cohort study found that elevated

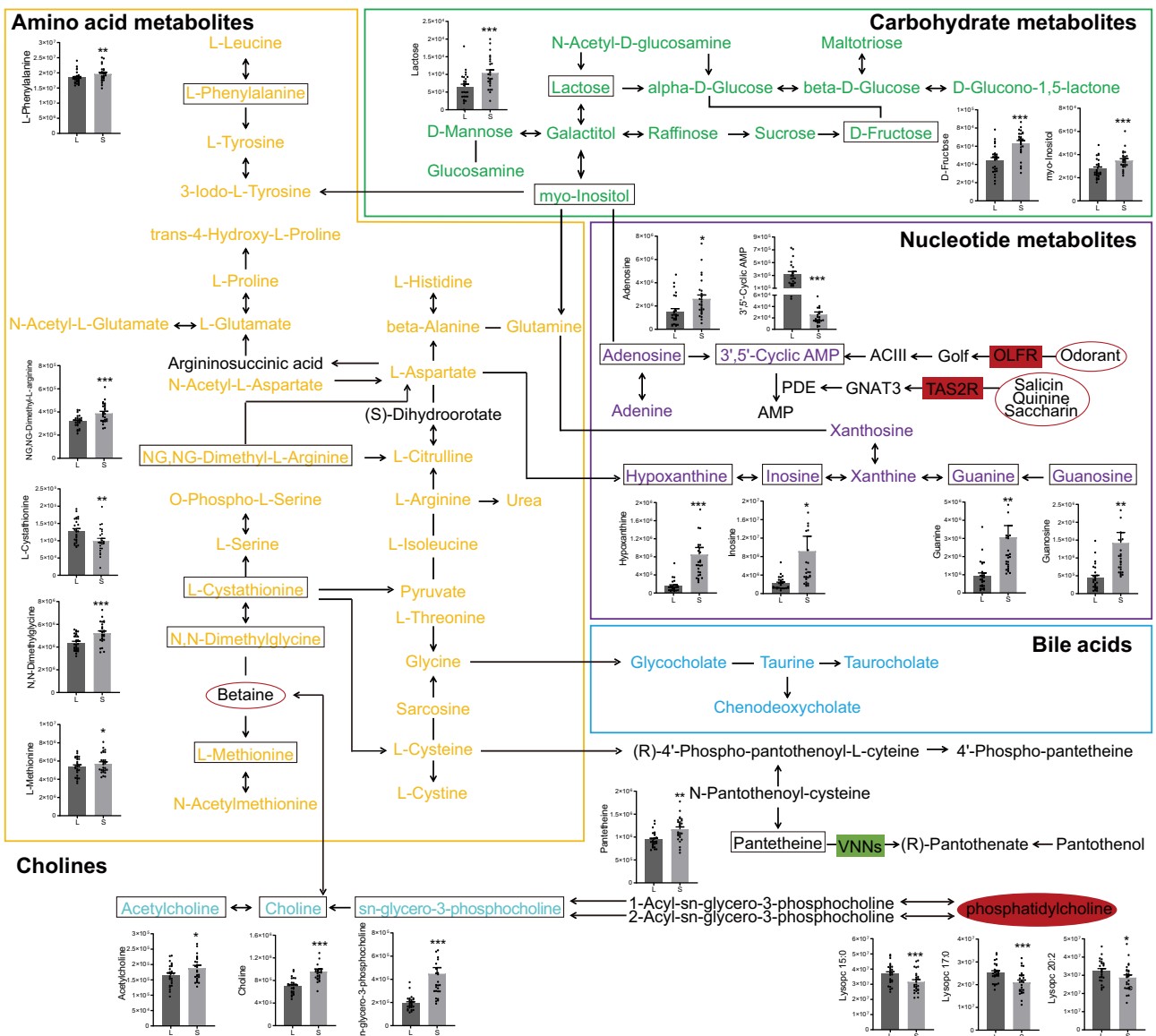

**Fig. 4 A schematic diagram of the complex inter-material pathway network.** It was formed by the direct interaction of differential metabolites and differential RNAs of the human follicular fluid from follicles of different sizes based on KEGG pathway map. The profiled metabolites are shown in the pathway network, and the significantly differential metabolites are presented as mean ± standard error of the mean (SEM). The gene colored red indicates increase in small follicles when compared with large follicles; the gene colored green indicates decrease in small follicles when compared with large follicles.

follicular-free fatty acid levels are associated with poor cumulus oocyte complex morphology[26]. Interestingly, differences in FF lipid composition in unexplained infertile patients occur across multiple lipid categories, harboring both positive and negative associations with lipid properties[27]. In obese women, the total lipid content of FF was markedly elevated, which caused lipotoxic impairment in oocyte maturation (OM) and early embryonic loss[23,28], while few phosphocholines exhibited reduced accumulation in poor ovarian responder patients[29]. Multiple studies reported that FF possessed a massive amount of phospholipid compounds, which were a major structural component of cellular and organelle membranes and may affect cellular interactions and contribute to fertility[30–32]. They include LPC, lysophosphatidic acid (LPA), sphingomyelin-1 phosphate, and sphingomyelin choline[33]. LPC can be metabolized to form LPA using phospholipase[34,35]. LPC is a good potential cytoplasmic messenger and serves as an activator of multiple secondary messengers, such as, extracellular signal-driven kinases and protein

kinase C[36,37]. A study revealed that the age-related alterations in the lipid metabolite LPC in FF may be related to oocyte quality[38]. In addition, a study examining IVF metabolomics demonstrated that LPC served a paracrine role on oocytes and participates in the acrosomal reaction[39]. Bertevello et al. analyzed lipid profiles of LFs and SFs within the same bovine ovaries, and reported that LPC, PC, and a number of non-annotated low-weight lipid species were differentially abundant between the follicles[25]. In our study, based on the global metabolic perspective, we screened out LPC which was differentially abundant in FF from human ovarian follicles of different stages, and there was a strong correlation between the concentrations of LPC in FF and blood serum at the same stage. Given these evidences, it indicates that LPC may play a crucial role in FD and OM in humans. However, the dose and mechanism of LPC still need to be studied.

Next, we attempted to correlate both FF and blood serum LPC with clinical epidemiological information, and found that LPC was negatively correlated with AFC, but significantly positively

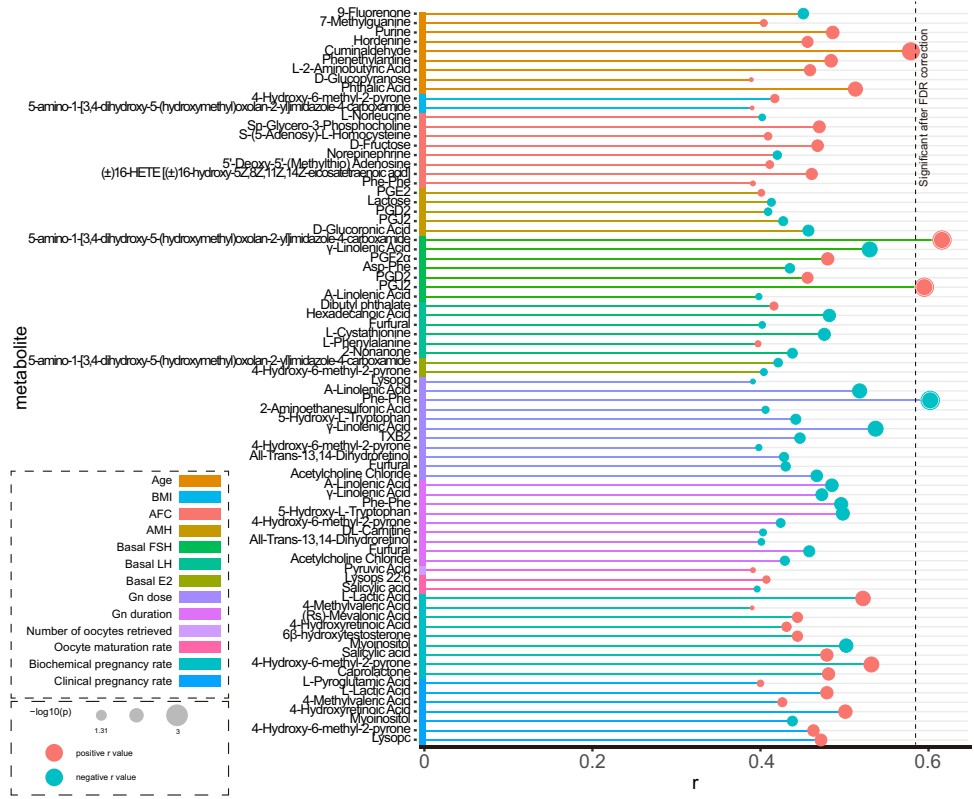

**Fig. 5 Significant spearman correlation between the metabolite ratios and clinical epidemiological information in follicular fluid samples.** The metabolite ratio = metabolite level in small follicles/metabolite level in large follicles. AFC antral follicle count, FSH follicle-stimulating hormone, AMH anti-mullerian hormone.

correlated with ovarian stimulation parameters FORT, FSI, and FOI. Furthermore, blood serum LPC can serve as a predictor of these three parameters. Ovarian sensitivity to gonadotropins varies from a patient to another, and plays a decisive role in the response of the ovary to stimulation. Therefore, it is a key element in improving IVF success rate[40–42]. FORT, FOI, and FSI can serve as a quantitative and qualitative marker of ovarian response, as they can effectively reflect the dynamic nature of FD, in response to exogenous Gn[43–46]. Additionally, studies revealed that FORT was discretely and negatively associated with serum AMH[47,48], while AMH and AFC were positively related to ovarian function. Therefore, the negative correlation between LPC and AFC and the positive correlation between LPC and FORT indicate that LPC is closely related to ovarian function. In addition to being a qualitative indicator of ovarian follicles competence, FORT also exhibits a significant correlation with clinical pregnancy outcome. Some studies revealed that a higher FORT was associated with improved pregnancy outcome[49–51]. However, in our study, LPC was not significantly associated with pregnancy outcome, which may be related to the various uncertain factors from FD to pregnancy, compromising the power to identify the association. Besides significant associations, insignificant difference in FORT values between pregnant and non-pregnant women was also reported[52]. Therefore, because LPC was strongly associated with FORT and FORT is a powerful tool for measuring ovarian reactivity, the association between LPC and pregnancy outcome still needs further investigation. Therefore, due to its association with multiple ovarian stimulation parameters, quantitative detection of LPC can successfully address the problem of ovarian sensitivity and thus become a more appropriate predictor of ovarian sensitivity.

In this study, we observed a negative correlation between BMI and LPC in FF, but no significant correlation between BMI and LPC in blood serum. Studies revealed that LPC was strongly associated with obesity. In particular, obese individuals have low plasma LPC concentrations[53–55], and there is indication that effective and long-term weight loss may restore LPC to normal levels[56]. In case of infancy and childhood overweight/obesity, LPC is strongly related to rapid growth. This suggests that the LPC levels may be a risk factor in some metabolic diseases such as PCOS, as studies have shown a decrease in LPC in obese PCOS patients[57], and may exert a metabolic effect on infant weight gain, thus elevating obesity risk[58]. Wallace et al. discovered low LPC content in obese populations may not be related to body weight, but to chronic inflammation[59]. Therefore, in obese individuals, the anti-inflammatory and cell-protective role of LPC is missing[60]. Further experimental studies are needed to investigate the interaction between LPC and obesity.

In this study, we found LPCs as key target mainly based on the conclusions of statistical and bioinformatics analysis (Figs. 3 and 4). The relationship between LPCs and various pathological conditions appears complex in previous papers[61–64]. The role of LPC in FF has not been clarified yet. Meanwhile, we also found other metabolites potentially related to reproduction in the differential metabolite list deserves concern, such as myoinositol. However, it is not the most centered metabolite of our comprehensive statistical and bioinformatics analysis, and its role in FF has been widely reported[65,66]. Therefore, considering our biomarker discovery strategy based on the statistical and biological considerations and novelty of the results, we chose LPC for further study[65–68].

Our study also revealed multi-omics layers information integration regarding FD in FF. Jiao et al. reported significant differences in mRNA expression profiles in FF from mature and immature ovarian follicles in healthy women[7]. We compared these differentially expressed mRNAs with our differential

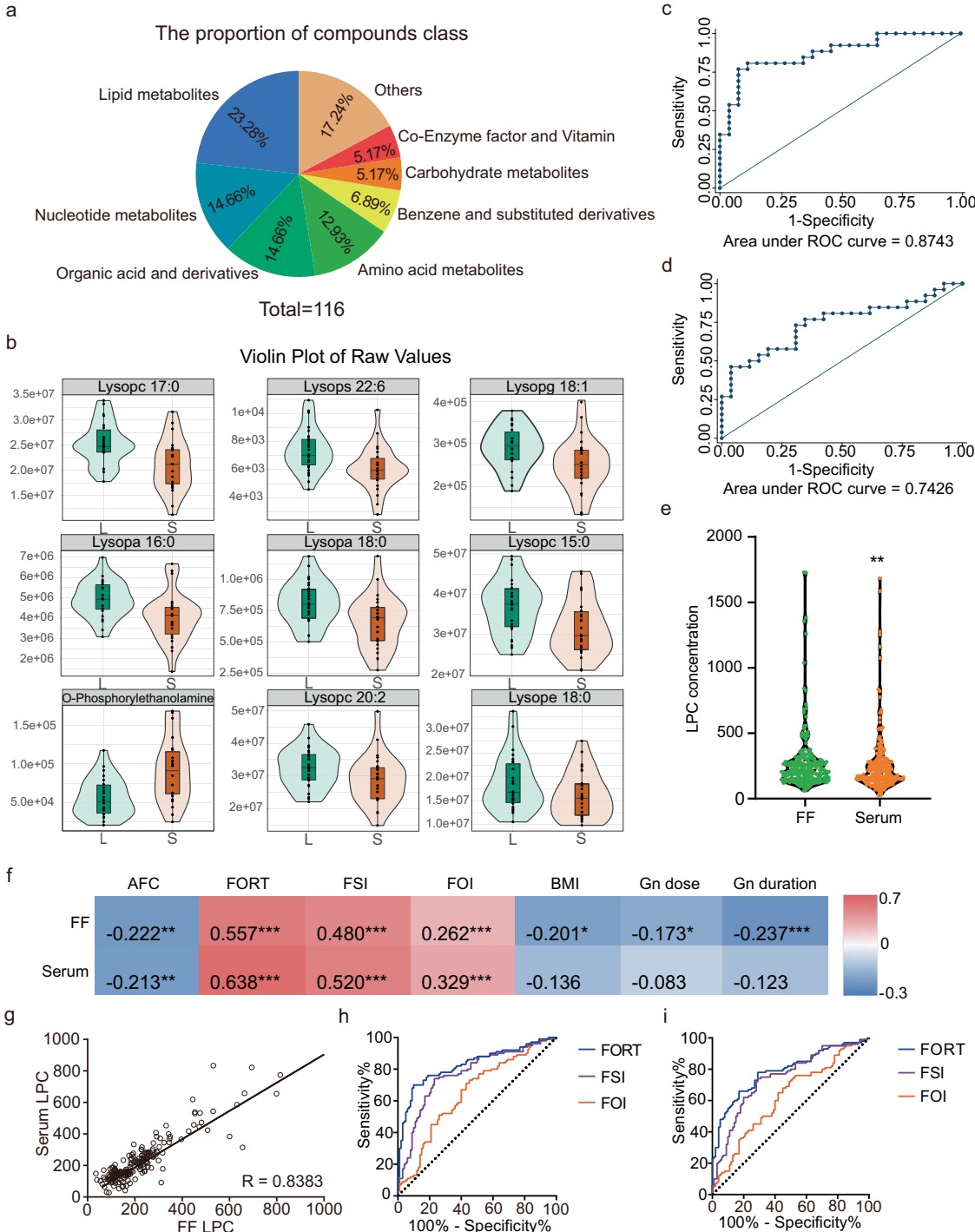

**Fig. 6 The lipid phospholipids in relation to follicular development in blood serum and follicular fluid. a** 116 differential metabolites divided into 8 classes, based on substance categories. **b** A violin diagram of nine lipid phospholipids showing data distribution and density. **c** ROC curves of the nine major lipid phospholipids found in differently sized follicles. **d** ROC curves of LPC in the follicular fluid of differently sized follicles. **e** Markedly different lysophosphatidylcholine (LPC) concentrations (pg/ml) in the follicular fluid and blood serum. **f** Relationship between LPC and epidemiological information, including BMI, AFC, ovarian stimulation parameters (Gn dose and duration), and ovarian sensitivity parameters (FORT, FSI, and FOI). **g** A strong linear relationship between follicular fluid LPC levels and blood serum LPC levels. **h** ROC curves for predicting ovarian sensitivity parameters (FORT, FSI, and FOI) using LPC levels in blood serum. **i** ROC curves for predicting ovarian sensitivity parameters (FORT, FSI, and FOI) using LPC levels in follicular fluid.

metabolite expression and found that VNNs, TAS2R, and OLFR were markedly correlated. In particular, OLFR gene suppression can lead to decreased sperm chemotaxis ability and inability to track oocyte for fertilization, thus reducing male fertility[69]. Moreover, in the female reproductive system, the olfactory pathway is mainly activated in the cumulus cells of competent

MII oocytes and plays critical roles in maturation, along with AC3 and OMP[70]. Meanwhile, the decrease of expression of VNNs in SFs which are pantetheinases that hydrolyse pantetheine to pantothenic acid was also included in the omics integration[71], indicating that the increase of pantetheine in SFs might be caused by the decreased hydrolysis due to the changed expression of

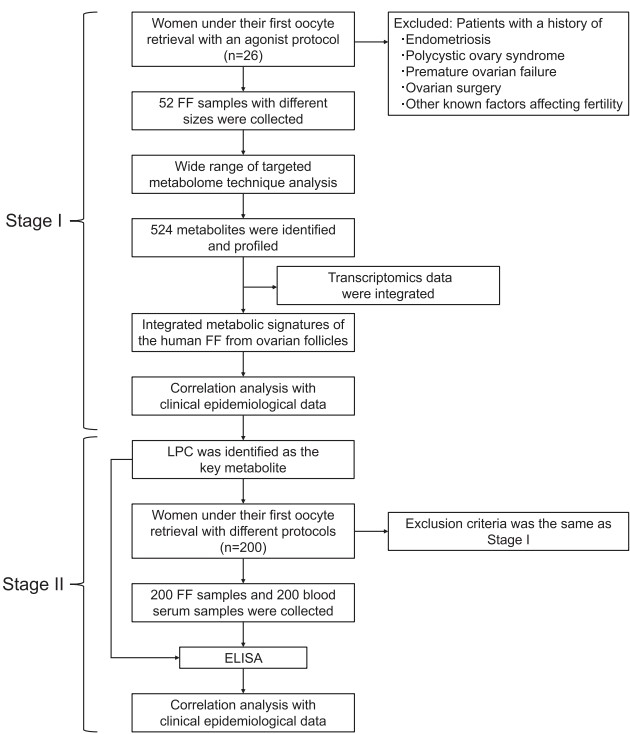

**Fig. 7 The two-stage study design of this study.** This was a two-stage experiment involving 452 samples from 226 individuals.

VNNs, thus providing novel insights into the mechanism underlying the metabolite changes.

In summary, we profiled FF metabolome during FD with the wide spectrum targeted metabolomics technique and integrated it with FF transcriptome, and demonstrated metabolic signatures in FF from ovarian follicles at different stages, filling the knowledge gap in humans. Among the differential metabolites, LPCs were the most notable changes. The validation of LPC in FF and blood serum in an independent population was further conducted, and it was found that total LPC, in both sample types (FF and blood serum) was closely related to ovarian stimulation parameters, suggesting that LPC can be used as an indicator of ovarian reactivity index, providing novel information regarding metabolites in relation to reproductive clinical indices in FF.

## Methods
### Stage I
*Study population.* The two-stage study design is shown in Fig. 7. We recruited women who underwent in vitro fertilization (IVF) during their initial cycle of ovarian stimulation, as part of an agonist protocol. Infertility in these women were brought on by either tubal complication or unknown cause of sterility. Unknown cause of sterility, in this research, was described as infertility even after two attempts of intrauterine insemination. These women had normal FD and none of the offered interventions affected FD, thus reducing errors, relative to other populations. We also selected women whose male partners were willing and able to provide healthy sperm for IVF. This was put in place to reduce effects of male compromising factors on the results. The patients excluded from the study were: patients with a history of endometriosis, polycystic ovary syndrome (PCOS), premature ovarian failure (POF), ovarian surgery, or other factors. All participants were recruited at the Reproductive Center of the Second Affiliated Hospital of Nanjing Medical University, China, and provided written informed consent. We also received ethical approval from the Second Affiliated Hospital of Nanjing Medical University.

*FF sample collection.* A total of 26 subjects were employed between April 2019 to November 2019 for participation in this study. Participants were pretreated with gonadotropin-releasing hormone agonist (GnRH-a) (triptorelin, Lizhu Pharmaceutical Trading Co., China), and started on Gn stimulation after confirmation of follicles down-regulation by transvaginal ultrasound and serum estradiol (E2) and luteinizing hormone measurements. Ultrasonography was the major tool

evaluating FD from Day 5 of stimulation until maturation. Once ≥1 follicle (s) achieved a diameter of 18 mm, we increased E2 levels to match the quantity of follicles. 10,000 IU human chorionic gonadotropin (HCG, Lizhu Pharmaceutical Trading Co., China) was provided 34–36 h before oocyte retrieval. Oocytes and their associated FF were obtained via transvaginal ultrasound-guided aspiration. FF samples from LFs (averaging 17–22 mm in diameter, which corresponded to the FF volume of 2.5–5.0 ml) and matched SFs (averaging 8–13 mm in diameter, which corresponded to the FF volume of 0.3–1.0 ml) were collected separately. The LFs were considered mature follicles and the corresponding SFs were immature follicles[7]. The FF samples next underwent centrifugation at $10,000 \times g$ for 10 min. The supernatant was retrieved and maintained at $-80\,°C$ for subsequent analysis.

*Metabolomic analysis.* The metabolomic data was analyzed by Wuhan Metware Biotechnology Co., Ltd, (Wuhan, 430070, China). This targeted metabolomics technique has been described in the previous report[72]. To conduct metabolomic assay, the FF sample was thawed on ice and 3 volumes of chilled methanol was introduced to 1 volume of FF sample, followed by a 3 min vortex and centrifugation at 12,000 rpm and $4\,°C$ for 10 min. The supernatant was centrifuged again at 12,000 rpm and $4\,°C$ for 5 min and the subsequent supernatant was used for liquid chromatography-tandem mass spectrometry (LC-MS/MS) analysis.

Sample extract analysis was done with an LC-electrospray ionization (ESI)-MS/MS system (UPLC, Shim-pack UFLC SHIMADZU CBM A system, https://www.shimadzu.com/; MS, QTRAP® 6500+ System, https://sciex.com/). The UPLC was equipped with Waters ACQUITY UPLC HSS T3 C18 (1.8 μm, 2.1 mm*100 mm) column with column temperature at $40\,°C$. The flow rate for UPLC was 0.4 mL/min with injection volume of 2 μL. The solvent system included water (0.04% acetic acid) and acetonitrile (0.04% acetic acid). The gradient program was 95:5 V/V at 0 min, 5:95 V/V at 11.0 min, 5:95 V/V at 12.0 min, 95:5 V/V at 12.1 min, and 95:5 V/V at 14.0 min. LIT and triple quadrupole (QQQ) scans were obtained on a triple quadrupole-linear ion trap mass spectrometer (QTRAP) accompanied by an ESI Turbo Ion-Spray interface, with both positive (5500 V,+ve) and negative ($-4500$ V, $-$ve) ion mode operations, and regulated by the Analyst 1.6.3 software (Sciex). The ESI source was operated with source temperature at $500\,°C$ and ion source gas I (GSI), gas II (GSII), curtain gas (CUR) at 55, 60, and 25.0 psi, respectively. The collision gas (CAD) was kept on high. Instrument tuning and mass calibration were carried out with 10 and 100 μmol/L polypropylene glycol solutions under QQQ and LIT modes, respectively. A particular group of MRM transitions were assessed in each period based on the eluted metabolites within the same time period. We also prepared quality control samples from a mixture of test samples and analyzed them to evaluate repeatability of samples under the same testing method. Additionally, during instrumental analysis, we analyzed a quality control (QC) sample along with every 10 FF samples to assess repeatability of this testing procedure.

### Stage II
*Study Population.* The participant inclusion and exclusion criteria were the same as Stage I, except for the ovarian stimulation protocol in the IVF cycle. In this part of the study, we examined controlled ovarian hyperstimulation (COH) programs by employing the GnRH-a long, GnRH-a short, and micro-stimulus protocols. Just like in stage I, we have received ethical approval from the Second Affiliated Hospital of Nanjing Medical University, and all participants provided written informed consent.

*FF and blood serum sample collection.* A total of 200 subjects were selected for analysis between January 2020 and August 2020. During this period, COVID-19 began to spread, but it did not affect the region where this study was conducted. Different COH programs were used according to individual conditions of the participants. However, the trigger criteria remained the same. FF samples from the first follicle of each participant were retrieved at the time of oocyte extraction. Blood samples were taken the day the oocytes were retrieved. The treatment of FF samples was consistent with IVF treatment in Stage I of this study, and blood samples underwent centrifugation at $3000 \times g$ for 10 min, and the supernatant serum samples were maintained at $-80\,°C$ for further analyses.

*Human LPC measurement.* The total LPC levels in FF and blood serum were measured with human LPC enzyme-linked immunoassay KIT (H2303, ELISA, Nanjing SenBeiJia Biological Technology Co., Ltd). The intra- and inter-assay coefficients of variations were 8 and 10%, respectively. The results were consistent between the metabolomics and ELISA measurements with correlation coefficient above 0.8 in our consistency test.

*Statistics and reproducibility.* Using the Metware database, the information and secondary spectrum data were assessed based on retention time (RT) and ions of identified metabolites. The metabolites were quantitatively analyzed with the multiple reaction monitoring (MRM) of triple quadrupole MS. The observed chromatographic peaks in varying samples of each metabolite were adjusted based on RT and peak type to ensure precision in our qualitative and quantitative analyses. Software Analyst 1.6.3 was employed for the processing of MS data. Univariate analysis was employed by using paired Student's t-test between two groups

and one-way ANOVA between three groups with SPSS version 13.0 (IBM Corp., Armonk, NY, USA). Multivariate analysis was conducted with principal component analysis (PCA) and orthogonal partial least squares discriminant analysis (OPLS-DA), and heatmap was built using "pheatmap" package with R software (version 3.6.0, https://www.r-project.org/). Hierarchical clustering analysis (HCA) was applied to evaluate the lineage relationship among samples. The pathway and network analysis of differential metabolites was conducted by using the "pathway analysis" module and "network analysis" module of MetaboAnalyst (version 5.0, https://www.metaboanalyst.ca/), respectively. The correlation between clinical data and metabolomic data was analyzed by spearman correlation analysis with SPSS (version 13.0), and false discovery rate (FDR) was applied for multiple comparisons. Based on the network establishment by MetScape plugin Cytoscape (version 3.6.1), the metabolic signatures were then discovered and visualized by multi-omics integration technology including our metabolomic data and published transcriptomic data of human FF with the same FD stage comparison settings as our current study[7]. Stata/SE (version 11.0) was used to perform receiver operator characteristic (ROC) curve analysis and to calculate the AUC, to estimate the predictive diagnostic potential of follicular development by using metabolomic biomarkers in FF. The results are expressed as the mean ± the standard error of the mean (SEM), and $p < 0.05$ was set as the significance threshold.

**Reporting summary.** Further information on research design is available in the Nature Research Reporting Summary linked to this article.

## Data availability
The metabolome datasets have been submitted to MetaboLights. The data that supports the findings of this study are available in the supplementary material of this article.

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

## Acknowledgements

Quantification of metabolite content in FF was performed by the wide range of targeted metabolome technique at Wuhan Metware Biotechnology Co. Ltd (Wuhan, China). This study was approved by the Second Affiliated Hospital of Nanjing Medical University. All patients provided written informed consent. This research was funded by the National Natural Science Foundation of China (Nos. 31900605 and 81971451), the Natural Science Foundation of Jiangsu Province (No. BK20190654), the Maternal and Child Health Scientific Research of Jiangsu Province (F202121), and the Science and Technology Development Foundation of Nanjing Medical University (No. NMUB2019052).

## Author contributions

J.H.Y., M.J.C., and Y.Q. planned and fashioned the experiments. J.H.Y. wrote, and M.J.C. revised and edited the manuscript. J.H.Y. conducted the experiments with others as follows: Y.B.L. and S.Y.L. performed clinical sample collection; Y.Z. aided with clinical epidemiological data retrieval and analysis; R.Z.F. and R.H. provided experimental assistance. The final manuscript was approved by all authors.

## Competing interests

The authors declare no competing interests.
