## [Peer Review File · Communications Biology]

Reviewers' comments:

Reviewer #1 (Remarks to the Author):

This paper just used targeted metabolomics to investigate the metabolic characteristics of human follicular fluid (FF), and whether there are metabolic predictors of follicular development (FD) with clinical implications. The study design is great and clear. However, the analysis is not enough to support the conclusions in this study. And the manuscript should be improved, especially the method section.

General comments:

This study's aim is to find the unknown biomarkers, so I am wondering why the authors select the targeted metabolomics, not untargeted?

The authors mentioned that they also measured the transcriptome data, however, I can't find any description of this in the method section.

Specific comments:

All the bar plots that show the mean values of repeat measures should be added with the dots.

Figure 2 a. It makes sense that OPLS-DA (supervised) can separate two groups. Please provide the unsupervised (PCA, HCA) and see the separation between the two groups.

For the biomarker discovery, the authors mentioned that they used the p value < 0.03 as a cutoff. I am wondering if those p values are adjusted and why 0.03 is selected as a cutoff.

Figure 2 E. The heatmap is used to show the two groups' samples. However, I can not get any information from this plot. Please cluster the rows and columns and see if the 37 metabolites can separate two groups of samples and if the metabolites are grouped by their classification.

Figure 2 F, again, are the p values for pathways adjusted?

Figure 2 G, what is the network? Is it a correlation network or a knowledge-based network? The authors mentioned that "Amino acids, nucleotides, and carbohydrates were the main metabolites in the network, and bile acids, choline as well as various LPCs were also included in the network." I am wondering how they defined "Amino acids, nucleotides, and carbohydrates were the main metabolites in the network"?

Line 94-98. I am not sure how the authors "integration of metabolomic and transcriptomic data." This is not clear and the conclusion is not confident from the analysis.

Figure 4. This plot is not good enough to show the correlations between the metabolite ratios and clinical epidemiological information. Please put the correlations on the x-axis and the dot size represents the p values. Again, please make sure that here the p values are adjusted.

For lines 99-112. The authors get some correlations between the metabolite ratios and clinical epidemiological information. Please clarify that how did you define the "significant correlations". And I can't get any conclusions from this analysis.

Figure 5 B, please add dots for each sample.

Figure 5 C-D, I can't find any descriptions that how the authors did this analysis in the main text or methods section.

Reviewer #2 (Remarks to the Author):

Yang et al from Chen and Qian labs/clinics provided an overall well-written manuscript and clear figures adequate of supporting the written text. The authors reported the use of metabolomics, transcriptomics and bioinformatics to arrive at FF and serum lysoPCs as a correlate of follicular development in humans. The use of two sets of individuals, one for discovery and the another for validation is a plus. While the analytical and statistical methodologies are sound and robust, this reviewer wonder if the authors maximize the data further in its interpretation. The data presented suggested other metabolites, primarily myoinositol, as a main target of downstream investigation and a missed opportunity not to follow up on it. It would therefore be imperative, and pertinent for the authors to explain why lysoPCs were chosen as a biomarker. Additionally, lysoPCs, as are PCs, have often been described in the literature for other diseases associated with inflammation.

Major points

-The purpose of biomarkers was not mentioned until the discussion. In this study, the need for a

biomarker was not clear and it leaves this reviewer uncertain of its purpose. The purpose should be specified in the introduction. The link between FF and serum and that a strong positive correlation of lysoPC in both compartment allowing potential use of it as a less invasive biomarker should be strengthened. However, it is still more important to state clearly why would a biomarker for follicular development be useful.

-Demographic information should already mention that the participants are IVF patients to set the appropriate context that these are individuals with fertility issues and seeking ART.

-Line 78-79. This reviewer would like to see the ranges, average and standard deviation of the study population. It would be interesting to see the range, and any notable large variation in the metabolite levels.

-Line 94, specify the LPCs.

The authors mentioned mechanistic study. What about mechanisms and for what purpose? Physiological, development or pathological or all of them.

-Line 112, Myoinositol showed negative correlation with pregnancy outcomes. This reviewer would like to assume that the grand objective of this study is to understand follicular development and how it affects reproduction. If this is true, myoinositol's correlation with pregnancy outcome would make it a prime metabolite for further studies. In addition, studies have shown that myoinositols have beneficial effects of PCOS patients, suggesting its levels and metabolism is related to fertility and therefore interesting to see if that holds true in this study of non-PCOS individuals. Ref, 10.1186/1477-7827-10-52 and 10.1186/s12958-021-00741-0.

-Line 113: It is interesting to see odd-chain lysoPCs. Confounding of ¹³C peaks of lysosphingomyelin is possible (see doi.org/10.4155/bio-2018-0036). Please provide the MS/MS evidence and also matched chemical standard retention times.

-Line 114: what are "substance" categories?

Line 126: from an independent population of 200 individuals would be more appropriate

-Line 130: please state what method was used for the total LPC measurements here, even though it was mentioned in M&M. Have the authors compared the analytical performances of ELISA versus LC-MS/MS for lysoPCs? Please provide information on how the two methods compare in terms of absolute and relative quantification.

-Line 130-131: Wouldn't comparing LPC between small and large follicles be the primary comparator in the Stage II validation study?

-Figure 2. Where is LysoPC and myoinositol in the figure panels? What are the VIP, fold change and pvalues of these metabolites? These are critical information to describe, in addition to the use GO and other network analysis, for the logic of why lysoPCs were given the focus for further investigation. Also, the data and literature suggest myoinositol as the metabolite to focus.

-Figures 2, 3 and 5. It should be "nucleotide metabolism" or "nucleotide metabolites" and for the other pathways.

-Table S4 and S5: This reviewer might have missed it but what is the statistical analysis performed?

-Clarify on the study population used in this manuscript and whether it overlaps with that of Study reference. 7. Is it the same population, and if so, were the authors "salami slicing"? The latter would be acceptable if the metabolomics were performed after the transcriptomics study which could very often happen. Please state so if it is what happened. If different, what did the authors do to 'match' the individuals?

-The use of animal studies for discussion and references would be less meaningful given that human data are presented here. Notably, aberrant lipid metabolism and levels in gynaecological diseases which have strong infertility component should be described (e.g. 10.1016/j.fertnstert.2018.05.003 and 10.1210/jc.2016-2692)

Minor points:

-Abstract: grammatical correction needed for abstract, line 18, wide range of targeted metabolomics.

-Figure 1 has too many panels, resulting in small figures which are difficult to read. It was especially difficult to read the metabolites.

-Line 106: elaborate Gn

-Table for MRM transitions of metabolites analysed. In the Acknowledgements, it was performed by a company but a list of metabolites analysed would be helpful.

Reviewer #3 (Remarks to the Author):

The paper by Yang et al. presents a quite interesting study on the identification of LPC in human follicular fluid as predictor of follicular development.

The study is developed in two stages. The first stage was based on 26 patients, whose follicular fluids were collected during ART procedures; for each patient, 2 samples of FF were collected, one from a small follicle (immature) and one from a large follicle (mature), with a total of 52 samples. Through an accurate metabolomic analysis, the authors identified 524 metabolites that differentiate small and large follicles and, among these metabolites, they managed to identify the metabolites that better differentiate mature and immature follicles. Using multi-omics integration technology the large number of identified differential metabolites and mRNAs transcriptomic data were directly connected. At the end of the first part of the study, the authors hypothesized that total LPC could be a good biomarker for further study and have a potential clinical use.

To verify this hypothesis, in the second stage of the study, 200 other patients were selected and, for each patient, 1 FF sample and 1 serum sample were collected. Total LPC was measured in both samples using enzyme-linked immunoassay. The new data confirmed that LPC is closely related to follicle development and ovarian response.

The study is not completely new in this field. A paper by Read et al. recently appeared on the correlation between the follicle diameter and FF metabolome profiles in lactating beef cows (Metabolites 2021, 11, 623 <https://doi.org/10.3390/metabo11090623>). This paper should be cited in the references.

However, the study by Yang et al is clearly designed and well-conducted. It may be of interest for those readers of Communication Biology working in the field of human reproductive health and metabolomics.

* One main issue: the authors enrolled 26 participants for the first stage of the study in the period from April 2019 to November 2019 when no COVID-19 was present, while the 200 women engaged for the second stage were enrolled in the period from January 2020 and August 2020 during COVID-19 pandemic. The authors did not mention any details on these last patients (i.e. if they were tested for SARS-CoV2 positivity before IVF treatments, if they received any vaccine,...). Did the authors consider this point? They are kindly requested to reply. Maybe, the authors should also write something in the paper concerning safety precautions, if any precaution was applied. Many papers appeared in the last two years about the necessity of strict patient screening and safety criteria in IVF laboratory (i.e. Rajput SK et al. 2021, Reprod Biomed Online. 42(6):1067-1074).

Minor revisions are required, as it follows:

Abstract:

- * The first sentence is not very clear and should be rephrased.
- *The number of participants in the first stage of the study should be inserted
- *The meaning of LPC abbreviation should be added.

Introduction

*Line 56 : "important metabolic signatures for further mechanistic study" it is not clear what is a "mechanistic study"

Results

*Line 92-93: "Most amino acids, nucleotides, carbohydrates and cholines levels elevated in SFs" should be changed in "Most levels of amino acids, nucleotides, carbohydrates and cholines were elevated in SFs"

*Line 125-126: "we collected samples from a 200 independent population" should be changed in "we collected samples from an independent population of 200 participants"

Reviewer #1 (Remarks to the Author):

This paper just used targeted metabolomics to investigate the metabolic characteristics of human follicular fluid (FF), and whether there are metabolic predictors of follicular development (FD) with clinical implications. The study design is great and clear. However, the analysis is not enough to support the conclusions in this study. And the manuscript should be improved, especially the method section.

General comments:

1. This study's aim is to find the unknown biomarkers, so I am wondering why the authors selected the targeted metabolomics, not untargeted?

Response: Thanks for the reviewer's comment. In the past years, both targeted and untargeted strategies have been applied to identify possible unknown biomarkers [22470063]. The widely targeted metabolomics database covers more than 1200 metabolites, including more than 160 amino acids and their derivatives, more than 220 organic acids and their derivatives, and more than 200 lipids, and so on. In order to detect metabolites qualitatively and quantitatively for further experiments, we chose wide spectrum targeted metabolomics. We have added these statements in the revised introduction.

2. The authors mentioned that they also measured the transcriptome data, however, I can't find any description of this in the method section.

Response: Thanks for the reviewer's comment. We integrated transcriptome data by referring to published data related to follicular development. This information has been described in the **Statistical analysis** section" the metabolic signatures were then discovered and visualized by multi-omics integration technology including our metabolomic data and published transcriptomic data of human FF with the same FD stage comparison settings as our current study".

Specific comments:

1) All the bar plots that show the mean values of repeat measures should be added with the dots.

Response: Thanks for the reviewer's comments. All these figures have been revised accordingly.

2) Figure 2 a. It makes sense that OPLS-DA (supervised) can separate two groups. Please provide the unsupervised (PCA, HCA) and see the separation between the two groups.

Response: Thanks for the reviewer's comments. The PCA model was made to distinguish the two groups, and it has been added to the Figure 2A. And the description of the analysis has been provided in the **Statistical analysis** section.

3) For the biomarker discovery, the authors mentioned that they used the p value < 0.03 as a cutoff. I am wondering if those p values are adjusted and why 0.03 is selected as a cutoff.

Response: Thanks for the reviewer's comments. Apologies for the confusion. We used $p < 0.05$ and fold change ≥ 2 and ≤ 0.5 of univariate analysis and $VIP > 1$ in the OPLS-DA model as a cutoff for the biomarker discovery. Due to the large number of differential

metabolites, metabolites with $p < 0.03$ and $VIP \geq 2$ were selected for heatmap display.

We added these explanations in **Metabolic signatures in FF from follicles during growth progression** section in the revised paper.

4) Figure 2 E. The heatmap is used to show the two groups' samples. However, I can not get any information from this plot. Please cluster the rows and columns and see if the 37 metabolites can separate two groups of samples and if the metabolites are grouped by their classification.

Response: Thanks for the reviewer's comments. This figure has been added to the supplemental figure 2.

5) Figure 2 F, again, are the p values for pathways adjusted?

Response: Thanks for the reviewer's comments. The p value was unadjusted. Moreover, we used multiple methods to verify the key metabolism involving in the follicular development (Figure 3B and C, Figure 4).

6) Figure 2 G, what is the network? Is it a correlation network or a knowledge-based network? The authors mentioned that "Amino acids, nucleotides, and carbohydrates were the main metabolites in the network, and bile acids, choline as well as various LPCs were also included in the network." I am wondering how they defined "Amino acids, nucleotides, and carbohydrates were the main metabolites in the network"?

Response: Thanks for the reviewer's comments. This figure was built by inputting both statistical and knowledge-based information using the Network module of Metaboanalyst (<https://www.metaboanalyst.ca/MetaboAnalyst/upload/MnetUploadView.xhtml>), which was described in the methods section. These metabolites were defined according to KEGG database, and its classification is colored in Figure 4. According to the number of differential metabolites in their classifications, amino acids, nucleotides and carbohydrates were the main changed classifications of metabolites in the network. The above information has been added in the **Metabolic signatures in FF from follicles during growth progression** section in the revised paper.

7) Line 94-98. I am not sure how the authors "integration of metabolomic and transcriptomic data." This is not clear and the conclusion is not confident from the analysis.

Response: Thanks for the reviewer's comments. The key metabolomic and transcriptomic changes were found by statistical analysis, and then they were connected by biological pathway presented in the KEGG database by using MetScape plugin Cytoscape (version 3.6.1). The integration method is according to our previous published paper [PMID: 33068521]. All the information has been involved in the **Statistical analysis** section in the revised paper and the biological information for omics integration has been added in the discussion.

8) Figure 4. This plot is not good enough to show the correlations between the metabolite ratios and clinical epidemiological information. Please put the correlations on the x-axis and the dot size represents the p values. Again, please make sure that here the p values are adjusted.

Response: Thanks for the reviewer's comments. The figure has been revised according

the reviewers' comment.

9) For lines 99-112. The authors get some correlations between the metabolite ratios and clinical epidemiological information. Please clarify that how did you define the "significant correlations". And I can't get any conclusions from this analysis.

Response: Thanks for the reviewer's comments. Since the follicular fluid samples of large follicles and small follicles were collected from the same participant, the ratios (metabolite concentrations in small follicles/the metabolite concentrations in large follicles) can reflect the state of this metabolite (normalized concentrations in small follicles after adjusting its concentrations in large follicles) in one individual. Through this analysis, we found that PGD2 and PGJ2 were found to be positively correlated with basal FSH and negatively correlated with AMH, which indicates that PGD2 and PGJ2 are two indicators reflecting ovarian function. And absence of associations between metabolites and both ovarian function and clinical pregnancy outcome, indicating that there is no direct correlation between ovarian function and clinical pregnancy outcome. Then after FDR correction, we found that the rates of two metabolites PGJ2 and 5-amino-1-[3,4-dihydroxy-5-(hydroxymethyl)oxolan-2-yl]imidazole-4-carboxamide were associated with basal FSH, and the rate of one metabolite Phe-Phe was associated with Gn dose. We added some context in the revised paper.

10) Figure 5 B, please add dots for each sample.

Response: Thanks for the reviewer's comments. This figure has been revised accordingly.

11) Figure 5 C-D, I can't find any descriptions that how the authors did this analysis in the main text or methods section.

Response: Thanks for the reviewer's comments. In Figure 6 C-D, we used Stata to perform receiver operator characteristic (ROC) curve analysis and to calculate the area under the curve (AUC), to estimate the predictive diagnostic potential of follicular development by using metabolomic biomarkers in follicular fluid. 9 differential metabolites, belonged to phospholipids, were combined to calculate the AUC in Figure 6C. And in Figure 6D, the AUC was calculated by combining the total LPC, including lysopc 15:0, lysopc 17:0 and lysopc 20:2. A description of the AUC has been provided in the **Methods** and **Results** sections.

Reviewer #2 (Remarks to the Author):

Yang et al from Chen and Qian labs/clinics provided an overall well-written manuscript and clear figures adequate of supporting the written text. The authors reported the use of metabolomics, transcriptomics and bioinformatics to arrive at FF and serum lysoPCs as a correlate of follicular development in humans. The use of two sets of individuals, one for discovery and the another for validation is a plus. While the analytical and statistical methodologies are sound and robust, this reviewer wonder if the authors maximize the data further in its interpretation. The data presented suggested other metabolites, primarily myoinositol, as a main target of downstream investigation and a missed opportunity not to follow up on it. It would therefore be imperative, and pertinent for the authors to explain why lysoPCs were chosen as a biomarker. Additionally, lysoPCs, as are PCs, have often been described in the literature for other diseases associated with inflammation.

Response: Thanks for the reviewer's comments. To find the key targets for further study is also our major concern. We found lysoPCs as key target mainly based on the conclusions of statistical and bioinformatics analysis, and the above results were detailed in Figure 3 and 4. This has been described in the **The LPCs in relation to FD and ovarian sensitivity in FF and blood serum** section. The relationship between LysoPCs and various pathological conditions appears complex in previous papers [PMID: 32599910, 31186025, 31793360, 30719830]. The role of lysoPCs in follicular fluid has not been clarified yet. Myoinositol is not the most centered metabolite of our comprehensive analysis of statistical and bioinformatics analysis, and its role in follicular fluid has been widely reported [PMID: 12042283, 27777587]. Therefore, considering the importance in the statistical and biological analysis and innovation of the results, we chose lysoPCs for further study.

-The purpose of biomarkers was not mentioned until the discussion. In this study, the need for a biomarker was not clear and it leaves this reviewer uncertain of its purpose. The purpose should be specified in the introduction. The link between FF and serum and that a strong positive correlation of lysoPC in both compartment allowing potential use of it as a less invasive biomarker should be strengthened. However, it is still more important to state clearly why would a biomarker for follicular development be useful.

Response: Thanks for the reviewer's comments. Metabolites related to follicular development found in follicular fluid may be used in culture medium to intervene the quality and development of oocyte, which will provide important information for improving the level of assisted reproductive technology in clinical. Meanwhile, studies have shown that follicular fluid vitamin D levels were consistent with serum vitamin D levels, and that serum vitamin D levels were positively correlated with normal fertilization rate [31319865]. So, further, if the link between follicular fluid and serum can be found and that a strong positive correlation of some metabolites in both compartments allowing potential use of it as a less invasive biomarker. Therefore, our study is aimed to find a biomarker that can reflect follicular development in blood through qualitative and quantitative analysis of human follicular fluid metabolome, which holds the promise for the novel biomarkers discovery for the diagnosis and treatment in the reproductive medicine. We have revised the **Introduction** accordingly.

-Demographic information should already mention that the participants are IVF patients to set the appropriate context that these are individuals with fertility issues and seeking ART.

Response: Thanks for the reviewer's comments. We have mentioned that the participants were IVF patients to set the appropriate context that these are individuals with fertility issues and seeking assisted reproductive technology in the revised **Demographic information** section.

-Line 78-79. This reviewer would like to see the ranges, average and standard deviation of the study population. It would be interesting to see the range, and any notable large variation in the metabolite levels.

Response: Thanks for the reviewer's comments. All the information has been provided in Table S3.

-Line 94, specify the LPCs.

Response: Thanks for the reviewer's comments. A total of 12 types of LPCs were detected in this study, including lysopc 14:0, 15:0, 16:0, 16:1, 17:0, 18:0, 18:1, 18:2, 18:3, 20:0, 20:1 and 20:2, among which lysopc 15:0, lysopc 17:0 and lysopc 20:2 had statistical differences. We have added details to the article.

-The authors mentioned mechanistic study. What about mechanisms and for what purpose? Physiological, development or pathological or all of them.

Response: Thanks for the reviewer's comments. We have added related statement in the revised paper.

This is of great clinical significance as it will aid in the selection of the important metabolic signatures for further mechanistic study for physiological and pathological research. Metabolites related to follicular development found in follicular fluid may be used in culture medium to intervene the quality and development of oocyte, which will provide important information for improving the level of assisted reproductive technology in clinical. Meanwhile, studies have shown that follicular fluid vitamin D levels were consistent with serum vitamin D levels, and that serum vitamin D levels were positively correlated with normal fertilization rate [31319865]. So, further, if the link between follicular fluid and serum can be found and that a strong positive correlation of some metabolites in both compartments allowing potential use of it as a less invasive biomarker, which holds the promise for both the novel biomarkers discovery for the diagnosis and therapeutic targets discovery for enhancing follicle and oocyte health in humans.

-Line 112, Myoinositol showed negative correlation with pregnancy outcomes. This reviewer would like to assume that the grand objective of this study is to understand follicular development and how it affects reproduction. If this is true, myoinositol's correlation with pregnancy outcome would make it a prime metabolite for further studies. In addition, studies have shown that myoinositols have beneficial effects of PCOS patients, suggesting its levels and metabolism is related to fertility and therefore interesting to see if that holds true in this study of non-PCOS individuals. Ref, 10.1186/1477-7827-10-52 and 10.1186/s12958-021-00741-0.

Response: Thanks for the reviewer's insightful comments. The main goal of this study was to understand follicular development and its impact on reproduction, so we examined metabolites at different stages of follicular development. We found lysoPCs mainly based on the conclusions of statistical and bioinformatics analysis, and the above results were detailed in Figure 2 and 3. The role of lysoPCs in follicular fluid has not been clarified yet. Myoinositol is not the most centered metabolite of our comprehensive analysis of statistical and bioinformatics analysis, and its role in follicular fluid has been widely reported [PMID: 12042283, 27777587]. Meanwhile, it was no longer significantly associated with biochemical pregnancy rate after FDR correction in the revised paper according to the reviewer's comment. Therefore, considering the importance in the statistical and biological analysis and innovation of the results, we chose lysoPCs for further study.

Meanwhile, we have related statement in the revised discussion. Given the biomarker discovery strategy in current study, we chose lysoPCs for further study. However, we also found other metabolites potentially related to reproduction in the differential metabolite list (Table S3) which needs further study in the future, such as myoinositol (Ref, 10.1186/1477-

7827-10-52 and 10.1186/s12958-021-00741-0).

-Line 113: It is interesting to see odd-chain lysoPCs. Confounding of ¹³C peaks of lysosphingomyelin is possible (see doi.org/10.4155/bio-2018-0036). Please provide the MS/MS evidence and also matched chemical standard retention times.

Response: Thanks for the reviewer's comments. The information can be seen in the table below. Odd-chain lysoPCs is usually ingested from the outside and can be detected in tissue or blood (<https://hmdb.ca/metabolites/HMDB0012108>; <https://hmdb.ca/metabolites/HMDB0010381>). According to the detection of LPC in large and small follicles by ELISA kit, it was found that the concentration of LPC in small follicles was significantly lower than that in large follicles, which confirmed the results of metabolomics about lysoPCs.

	Precursor	Quantification Ion	RT (min)
Lysopc 14:0	468.3	184	7.54
Lysopc 15:0	482.3	184.1	8.26
Lysopc 16:0	496.33	184	8.69
Lysopc 16:1	494.3147	184.0737	8.06
Lysopc 17:0	510.1	184.4	9.34
Lysopc 18:0	524.3629	184.0737	9.68
Lysopc 18:1	522.3465	184.0737	9.09
Lysopc 18:2	520.3302	184.0737	8.42
Lysopc 18:3	518.3162	184.0737	8.77
Lysopc 20:0	552.4029	184.0737	10.89
Lysopc 20:1	550.37	184	10.15
Lysopc 20:2	548.3624	184	9.38

-Line 114: what are "substance" categories?

Response: Thanks for the reviewer's comments. It is based on KEGG database. We classified 116 differential metabolites into amino acid metabolites, nucleotide metabolites and lipid metabolites, etc. We have revised the sentence accordingly.

-Line 126: from an independent population of 200 individuals would be more appropriate.

Response: Thanks for the reviewer's comments. We have revised this sentence.

-Line 130: please state what method was used for the total LPC measurements here, even though it was mentioned in M&M. Have the authors compared the analytical performances

of ELISA versus LC-MS/MS for lysoPCs? Please provide information on how the two methods compare in terms of absolute and relative quantification.

Response: Thanks for the reviewer's comments. We measured the total LPC levels with human LPC enzyme-linked immunoassay KIT (H2303, ELISA, Nanjing SenBeiJia Biological Technology Co., Ltd). This is a mature kit for scientific research. We previously tested the same substance using metabolomics and ELISA, and the results were consistent between the metabolomics and ELISA measurements ($R = 0.8054$).

-Line 130-131: Wouldn't comparing LPC between small and large follicles be the primary comparator in the Stage II validation study?

Response: Thanks for the reviewer's comments. The comparison of LPC between small and large follicles was to find biomarkers related to follicular development. The stage II experiment is to study the extrapolation of key metabolites in different body fluids, such as follicular fluid and blood serum. Meanwhile, in this stage, we examined the association between LPC and ovarian sensitivity parameters which is also related to follicular development, to verify previous findings.

-Figure 2. Where is LysoPC and myoinositol in the figure panels? What are the VIP, fold change and pvalues of these metabolites? These are critical information to describe, in addition to the use GO and other network analysis, for the logic of why lysoPCs were given the focus for further investigation. Also, the data and literature suggest myoinositol as the metabolite to focus.

Response: Thanks for the reviewer's comments. LysoPC and myoinositol are not shown in the figure panels, because Figure 2E showed the top 40 metabolites with $VIP \geq 2$, and Figure 3A showed the 37 metabolites with $p < 0.03$ and $VIP \geq 2$. The following table shows the p values, VIP and fold-change of metabolites myoinositol and LysoPCs. We found lysoPCs mainly based on the conclusions of statistical and bioinformatics analysis, and the above results were detailed in revised Figure 3B, C and 4 from the pathway and network analysis. This has been described in the **The LPCs in relation to FD and ovarian sensitivity in FF and blood serum** section. The relationship between LysoPCs and various pathological conditions appears complex in previous papers [PMID: 32599910, 31186025, 31793360, 30719830]. The role of lysoPCs in follicular fluid has not been clarified yet. Myoinositol is not the most centered metabolite of our comprehensive analysis of statistical and bioinformatics analysis, and its role in follicular fluid has been widely reported [PMID: 12042283, 27777587]. Therefore, considering the importance in the statistical and biological analysis and innovation of the results, we chose lysoPCs for further study. The detailed information has been provided in Table S3.

	P	VIP	Fold-change
Myoinositol	0.008117	1.393226	1.205938
Lysopc 14:0	0.010469	0.92816	0.89804
Lysopc 15:0	0.000833	1.560547	0.858443
Lysopc 16:0	0.164322	1.204841	0.982161
Lysopc 16:1	0.081143	0.958819	0.93408
Lysopc 17:0	4.19E-05	1.809915	0.832882
Lysopc 18:0	0.170795	1.23469	0.973795
Lysopc 18:1	0.762888	0.099948	1.007044
Lysopc 18:2	0.385648	0.75801	0.985869
Lysopc 18:3	0.28066	0.21129	1.082193
Lysopc 20:0	0.181777	0.855913	0.88789
Lysopc 20:1	0.014247	0.806677	0.881358
Lysopc 20:2	0.011727	1.33147	0.885049

-Figures 2, 3 and 5. It should be “nucleotide metabolism” or “nucleotide metabolites” and for the other pathways.

Response: Thanks for the reviewer’s comments. Apologies for the mistake. We have corrected the statement.

-Table S4 and S5: This reviewer might have missed it but what is the statistical analysis performed?

Response: Thanks for the reviewer’s comments. In Table S6 and S7, we used one-way ANOVA to analyze differences among three groups divided by tertiles. This information has been involved in the **Statistical analysis** section and indicated in the table notes.

-Clarify on the study population used in this manuscript and whether it overlaps with that of Study reference. 7. Is it the same population, and if so, were the authors “salami slicing”? The latter would be acceptable if the metabolomics were performed after the transcriptomics study which could very often happen. Please state so if it is what happened. If different, what did the authors do to ‘match’ the individuals?

Response: Thanks for the reviewer’s comments. The population in this study was not the same population as in study of reference 7. In this study, we integrated the differential transcripts information based on the comparison between large and small follicles using follicular fluid in reference 7 with differential metabolites information based on the comparison between large and small follicles using follicular fluid with the same study design in our study. The key metabolomic and transcriptomic changes were found by statistical analysis, and then they were connected by biological pathway presented in the KEGG database by using MetScape plugin Cytoscape (version 3.6.1). The integration method is according to our previous published paper [PMID: 33068521]. All the information has been involved in the **Statistical analysis** section.

-The use of animal studies for discussion and references would be less meaningful given that human data are presented here. Notably, aberrant lipid metabolism and levels in gynaecological diseases which have strong infertility component should be described (e.g. 10.1016/j.fertnstert.2018.05.003 and 10.1210/jc.2016-2692)

Response: Thanks for the reviewer's comments. Multiple studies demonstrated that the human ovarian follicular environment surrounding the oocyte exhibits a unique metabolite profile, with distinct localisation of lipids within follicular fluid and oocytes, and maturation in an environment with altered lipid content may be detrimental to oocytes. We have involved the indicated reference and more human evidence.

Minor points:

-Abstract: grammatical correction needed for abstract, line 18, wide range of targeted metabolomics.

Response: Thanks for the reviewer's comments. The grammatical correction has been conducted.

-Figure 1 has too many panels, resulting in small figures which are difficult to read. It was especially difficult to read the metabolites.

Response: Thanks for the reviewer's comments. For ease of reading, we have split Figure 2 into two graphs, namely, Figure 2 and Figure 3.

-Line 106: elaborate Gn

Response: Thanks for the reviewer's comments. Gn has been elaborated in the revised paper.

-Table for MRM transitions of metabolites analysed. In the Acknowledgements, it was performed by a company but a list of metabolites analysed would be helpful.

Response: Thanks for the reviewer's comments. We have added this information in Table S3. The key metabolite (lysoPCs) transitions have been provided in the response to the above comment.

Reviewer #3 (Remarks to the Author):

The paper by Yang et al. presents a quite interesting study on the identification of LPC in human follicular fluid as predictor of follicular development.

The study is developed in two stages. The first stage was based on 26 patients, whose follicular fluids were collected during ART procedures; for each patient, 2 samples of FF were collected, one from a small follicle (immature) and one from a large follicle (mature), with a total of 52 samples. Through an accurate metabolomic analysis, the authors identified 524 metabolites that differentiate small and large follicles and, among these metabolites, they managed to identify the metabolites that better differentiate mature and immature follicles. Using multi-omics integration technology the large number of identified differential metabolites and mRNAs transcriptomic data were directly connected. At the end of the first part of the study, the authors hypothesized that total LPC could be a good biomarker for further study and have a potential clinical use.

To verify this hypothesis, in the second stage of the study, 200 other patients were selected and, for each patient, 1 FF sample and 1 serum sample were collected. Total LPC was measured in both samples using enzyme-linked immunoassay. The new data confirmed that LPC is closely related to follicle development and ovarian response.

The study is not completely new in this field. A paper by Read et al. recently appeared on the correlation between the follicle diameter and FF metabolome profiles in lactating beef

cows (Metabolites 2021, 11, 623 <https://doi.org/10.3390/metabo11090623>). This paper should be cited in the references.

However, the study by Yang et al is clearly designed and well-conducted. It may be of interest for those readers of Communication Biology working in the field of human reproductive health and metabolomics.

Response: Thanks for the reviewer's comments. This paper in lactating beef cows has some relevance to our research in humans and has been cited in the references.

* One main issue: the authors enrolled 26 participants for the first stage of the study in the period from April 2019 to November 2019 when no COVID-19 was present, while the 200 women engaged for the second stage were enrolled in the period from January 2020 and August 2020 during COVID-19 pandemic. The authors did not mention any details on these last patients (i.e. if they were tested for SARS-CoV2 positivity before IVF treatments, if they received any vaccine...). Did the authors consider this point? They are kindly requested to reply. Maybe, the authors should also write something in the paper concerning safety precautions, if any precaution was applied. Many papers appeared in the last two years about the necessity of strict patient screening and safety criteria in IVF laboratory (i.e. Rajput SK et al. 2021, Reprod Biomed Online. 42(6):1067-1074).

Response: Thanks for the reviewer's comments. Although COVID-19 pandemic began in 2020, the 200 participants were enrolled from January 2020 and August 2020, and not been infected with the virus. None of them received the vaccine either, as large-scale vaccination began in the region around December 2021. We added the information in the **Methods** section of the revised paper.

Minor revisions are required, as it follows:

Abstract:

* The first sentence is not very clear and should be rephrased.

Response: Thanks for the reviewer's comments. It has been corrected in the article.

*The number of participants in the first stage of the study should be inserted

Response: Thanks for the reviewer's comments. It has been added in the article.

*The meaning of LPC abbreviation should be added.

Response: Thanks for the reviewer's comments. It has been added in the article.

Introduction

*Line 56: "important metabolic signatures for further mechanistic study" it is not clear what is a "mechanistic study"

Response: Thanks for the reviewer's comments. The sentence has been revised accordingly.

Results

*Line 92-93: "Most amino acids, nucleotides, carbohydrates and cholines levels elevated in SFs" should be changed in "Most levels of amino acids, nucleotides, carbohydrates and cholines were elevated in SFs"

Response: Thanks for the reviewer's comments. It has been corrected in the revised paper.

*Line 125-126: "we collected samples from a 200 independent population" should be

changed in “we collected samples from an independent population of 200 participants”

Response: Thanks for the reviewer’s comments. It has been corrected in the revised paper.

Reviewers' comments:

Reviewer #1 (Remarks to the Author):

Thanks for the authors' detailed response to the comments and concerns. Most of my comments have been addressed. Only question about the marker selection. The authors mentioned that they used the $p < 0.03$ and other criteria to define the bio markers, so my think they didn't adjust the p values, which may have lots of fault positive markers in their results. Could you explain that?

Reviewer #2 (Remarks to the Author):

- "LPC" in the title should be fully spelled out.

- The first paragraph in Introduction seems abrupt, and this reviewer's suggestion is to remove it entirely.

- Line 95, $p < 0.03$ typo is still uncorrected.

- Line 111, lysopc should be written as lysoPC.

- Previously, I asked about myoinositol. The authors provided a satisfactory reply in the rebuttal letter and noted amendments were made in the main text. However, in the revised manuscript, I do not see the amendments. The authors should directly include the line number in the main text for easy reference. It is important to discuss why myoinositol was not investigated further, and instead lysoPC was the focus of the study.

- The analytical comparison for measuring total lysoPC using ELISA vs LC-MS/MS should be added in Methods to gain confidence that the measurements can be compared between the two methods.

A minor revision will be appropriate.

Reviewer #3 (Remarks to the Author):

The authors have satisfied all the changes I had required.
I think that the revised manuscript can be accepted in the present version

Reviewers' comments:

Reviewer #1 (Remarks to the Author):

Thanks for the authors' detailed response to the comments and concerns. Most of my comments have been addressed. Only question about the marker selection. The authors mentioned that they used the $p < 0.03$ and other criteria to define the bio markers, so my think they didn't adjust the p values, which may have lots of fault positive markers in their results. Could you explain that?

Response: Thanks for the reviewer's comments. The confidence of the markers is also our concern. Therefore, metabolites with fold change ≥ 2 or ≤ 0.5 , $VIP > 1$, and $p < 0.05$ were considered as differential metabolites in our study, which is a common criterion to find differential metabolites and control false positive findings in metabolomics researches [PMID: 34915369, 35660575, 35037431, 35399654]. The statement has been revised and reference has been added in the **Metabolic signatures in FF from follicles during growth progression** section. Moreover, the criteria for selecting differential metabolites as potential biomarkers were next based on the combination of statistical and biological considerations to further improve the confidence of the results (Figure 3 and 4).

Reviewer #2 (Remarks to the Author):

- "LPC" in the title should be fully spelled out.

Response: Thanks for the reviewer's comments. It has been modified in the article.

- The first paragraph in Introduction seems abrupt, and this reviewer's suggestion is to remove it entirely.

Response: Thanks for the reviewer's comments. We have removed the first paragraph in **Introduction** entirety.

- Line 95, $p < 0.03$ typo is still uncorrected.

Response: Thanks for the reviewer's comments. A total of 116 significantly regulated metabolites were determined by $p < 0.05$, $VIP > 1$, and fold change ≥ 2 or ≤ 0.5 . Due to the large number of differential metabolites, 37 metabolites with $p < 0.03$ and $VIP \geq 2$ were selected for heatmap display for clarity. We have revised this sentence to make this point clear accordingly.

- Line 111, lysopc should be written as lysoPC.

Response: Thanks for the reviewer's comments. It has been corrected in the article.

- Previously, I asked about myoinositol. The authors provided a satisfactory reply in the rebuttal letter and noted amendments were made in the main text. However, in the revised manuscript, I do not see the amendments. The authors should directly include the line number in the main text for easy reference. It is important to discuss why myoinositol was not investigated further, and instead lysoPC was the focus of the study.

Response: Thanks for the reviewer's comments. The myoinositol related amendments have been added to the article **Discussion** (Line 275-284).

- The analytical comparison for measuring total lysoPC using ELISA vs LC-MS/MS should be added in Methods to gain confidence that the measurements can be compared between the two methods.

Response: Thanks for the reviewer's comments. We have added related statement in **Human LPC measurement** section of **Methods** part.

A minor revision will be appropriate.

Response: Thanks for the reviewer's comments. We have revised the paper point by point according to all your comments. We deeply hope our response is satisfactory.

Reviewer #3 (Remarks to the Author):

The authors have satisfied all the changes I had required.

I think that the revised manuscript can be accepted in the present version.

Response: Thanks for the reviewer's comments.